Ankyrin domains across the Tree of Life

Jernigan Kristin K. 1
Bordenstein Seth R. 1 2 s.bordenstein@vanderbilt.edu
1 Department of Biological Sciences, Vanderbilt University , Nashville , Tennessee , United States of America
2 Department of Pathology, Microbiology, and Immunology, Vanderbilt University , Nashville , Tennessee , United States of America
Weightman Andrew
Electronic publication date: 2014 Feb 6
Publication date: 2014
Volume: 2
Electronic Location ID: e264
Received 2013 Aug 28; Accepted 2014 Jan 15
Copyright: © 2014 Jernigan et al.
Copyright year: 2014
Copyright holder: Jernigan et al.
License: This is an open access article distributed under the terms of the Creative Commons Attribution License, which permits unrestricted use, distribution, and reproduction in any medium, provided the original author and source are credited.
License URL: https://creativecommons.org/licenses/by/3.0/

Keywords: Protein domains, Symbiosis, Archaea, Bacteria, Ankyrin repeat domains

Funding: This research was made possible by NIH awards F32 GM 100778 and 5T32HD007043-34 to KKJ, and R01 GM085163 to S.R.B. The funders had no role in study design, data collection and analysis, decision to publish, or preparation of the manuscript.

==============================
Ankyrin (ANK) repeats are one of the most common amino acid sequence motifs that mediate interactions between proteins of myriad sizes, shapes and functions. We assess their widespread abundance in Bacteria and Archaea for the first time and demonstrate in Bacteria that lifestyle, rather than phylogenetic history, is a predictor of ANK repeat abundance. Unrelated organisms that forge facultative and obligate symbioses with eukaryotes show enrichment for ANK repeats in comparison to free-living bacteria. The reduced genomes of obligate intracellular bacteria remarkably contain a higher fraction of ANK repeat proteins than other lifestyles, and the number of ANK repeats in each protein is augmented in comparison to other bacteria. Taken together, these results reevaluate the concept that ANK repeats are signature features of eukaryotic proteins and support the hypothesis that intracellular bacteria broadly employ ANK repeats for structure-function relationships with the eukaryotic host cell.

Introduction

Ankyrin (ANK) repeats are ubiquitous structural motifs in eukaryotic proteins. They function as scaffolds to facilitate protein-protein interactions involved in signal transduction, cell cycle regulation, vesicular trafficking, inflammatory response, cytoskeleton integrity, transcriptional regulation, among others (Mosavi et al., 2004). Consistent with the necessity of their function, amino acid substitutions in the ANK repeats of a protein (ANK-containing proteins) are associated with a number of human diseases including cancer (p16 protein) (Tang, Fersht & Itzhaki, 2003), neurological disorders such as CADASIL (Notch protein) (Joutel et al., 1996), and skeletal dysplasias (TRPV4 protein) (Inada et al., 2012; Mosavi et al., 2004). In addition, variations in the amino acid sequence of the human ANKK1 are associated with addictive behaviors such as alcoholism and nicotine addiction (Ponce et al., 2008; Suraj Singh, Ghosh & Saraswathy, 2013).

The structure of each individual 33 amino acid ANK repeat begins with a β-turn that precedes two antiparallel α-helices and ends with a loop that feeds into the next repeat. These interconnected protein motifs stack one upon another to form an ANK domain (Gorina & Pavletich, 1996; Sedgwick & Smerdon, 1999). The prevalence and varied functionality of ANK-containing proteins in eukaryotes can be attributed to (i) the strong degeneracy of the 33 amino acid repeat that allows for the specificity of individual molecular interactions, and (ii) the variability in the number of individual repeats in an ANK domain, which provides a platform for protein interactions (Li, Mahajan & Tsai, 2006; Sedgwick & Smerdon, 1999).

Because the ANK repeat was discovered in Saccharomyces cerevisiae, Schizosaccharomyces pombe, and Drosophila melanogaster (Breeden & Nasmyth, 1987), they were quickly prescribed as a signature feature of eukaryotic proteins. Despite the conventional wisdom (until recently) and frequent citations in the literature that ANK repeats are taxonomically restricted to eukaryotes, there has been no systematic investigation to assess their distribution across the diversity of life. Several related questions on the comparative biology of ANK repeats can be addressed: Are ANK-containing proteins more prevalent in the domains Eukarya than Bacteria and Archaea and to what extent? What is the typical fraction of a proteome dedicated to proteins containing ANK repeats across the three domains of life? Are ANK-containing proteins distributed non-randomly with respect to taxonomy or lifestyle?

In this study, we establish a new threshold on the distribution of ANK repeats across the tree of life. Further, the enrichment of ANK-containing proteins in symbiotic bacteria provides comprehensive support to experimental cases in which ANK-containing proteins promote interactions between bacterial and eukaryotic cells.

Materials and methods

ANK data acquisition and analysis

All genome information was obtained from the SUPERFAMILY v1.75 database (SUPERFAMILY; Wilson et al., 2009), including the taxonomy, and number of ANK-containing proteins (Table S1). The SUPERFAMILY database currently contains protein domain information on 2,489 strains, where there can be more than one strain representing a single phylogenetic species. This database is an archive of structural and functional domains in proteins of sequenced genomes (Wilson et al., 2009), which are annotated using hidden Markov models through the SCOP (Structural Classification of Proteins) SUPERFAMILY protein domain classification (Gough et al., 2001; SUPERFAMILY). We note appropriate caution that ANK-containing proteins are identified based on a computational framework and are not experimentally confirmed. We used NCBI’s Genome resource (NCBI Genome resource) to obtain total gene and protein numbers for each strain in the analysis. To determine the percent of a strain’s total protein number (proteome) that is composed of ANK-containing proteins, the number of ANK-containing proteins was divided by the total number of proteins and multiplied by a factor of 100. Only strains with available total protein information were used in this analysis. For the bacterial class and lifestyle analysis, an average of the number and/or percent of ANK-containing proteins for all strains of the same species were used for these analyses. For the lifestyle analysis, ANK-containing protein information on Cardinium hertigii was added to the analysis because detailed information regarding its ANK-containing proteins was recently published (Penz et al., 2012).

To analyze the amino acid sequence of ANK repeats and generate the consensus sequence for Archaea, we obtained the sequence ID of ANK-containing proteins from SUPERFAMILY v1.75 (SUPERFAMILY) and the amino acid sequence from NCBI’s Proteins database (NCBI Protein resource). We used SMART (Letunic, Doerks & Bork, 2012) to identify the number and location of each individual ANK repeat in the protein (Letunic, Doerks & Bork, 2012; Schultz et al., 1998). For the amino acid sequence identity analysis, individual ANK repeat sequences were aligned using MUSCLE using default parameters (Edgar, 2004) and the percent identity of the sequences was calculated in Geneious Pro 5.6.2 (Biomatters, 2010). To generate the archaeal ANK consensus sequence, all 132 ANK repeat sequences from the ANK-containing proteins identified in the SUPERFAMILY database were utilized. To generate the eukaryotic ANK consensus sequence, ANK repeat sequences from one ANK-containing protein from each phylum was identified in the SUPERFAMILY database, the ANK repeat was identified by SMART and utilized, resulting in a total of 153 ANK-repeat sequences (Table S2). When comparing ANK repeat sequences from two strains, the average of all combinations of ANK repeat comparisons was used. For the eukaryotic and archaeal consensus sequence, all indels and ends were trimmed after the ANK repeats were aligned by MUSCLE. The consensus sequence was generated by Geneious.

16S rRNA phylogenetic tree and independence analysis

We selected one representative 16S rRNA sequence from each bacterial class and aligned them by MUSCLE in Geneious Pro 5.6.2 (Table S3). This alignment was then used to reconstruct a phylogenetic tree that reflects the well-established ancestry of the bacterial classes for a phylogenetic independence test of the abundance of ANK-containing proteins. Prior to building the tree, a DNA substitution model for the alignment was selected by using jModelTest, version 2.1.3 using default parameters (Darriba et al., 2012; Guindon & Gascuel, 2003) A Bayesian phylogenetic tree was generated by MrBayes using the HKY85 IG model of DNA sequence evolution using default parameters (Huelsenbeck & Ronquist, 2001; Ronquist & Huelsenbeck, 2003; Hasegawa, Kishino & Yano, 1985). For testing phylogenetic independence of ANK-containing proteins, the PDAP program in Mesquite vs 2.75 was used to generate independent contrasts for the data in Fig. 3B using default parameters (Maddison & Maddison, 2006; Midford, Garland & Maddison, 2005) Phylogenetic Independence version 2.0 (Reeve & Abouheif, 2003) performed the Test For Serial Independence (TFSI) using default parameters based on the Bayesian tree.

Results

ANK-containing proteins across the tree of life

The consensus amino acid sequences for the ANK repeats in each domain of life are shown in Fig. 1 (Al-Khodor et al., 2010; Mosavi et al., 2004) (Table S2). There is a notable correspondence in amino acid identity and similarity across the domains, with the highest values between Eukarya and Bacteria (76.7% identity), followed by Archaea and Bacteria (73.3% identity), and then Eukarya and Archaea (66% identity). Despite the conservation of the domain-specific consensus sequences, there can be substantial amino acid sequence diversity at each position of the ANK repeat. For example, this variation is evident in the Archaea, where the mean % of the sequences ± standard deviation that establishes each consensus amino acid is 49.6 ± 24.7% (Table S4). Indeed, seven amino acid positions form a consensus from less than one quarter of the sequences.

Figure 1 ANK repeat consensus sequence across all domains of life.

Comparison of consensus sequences previously derived from (i) 153 Eukarya ANK repeat sequences (Table S2), (ii) 132 Archaea ANK repeat sequences and (iii) Bacteria ANK repeat sequences (Al-Khodor et al., 2010). The amino acid color scheme indicates that the amino acids share similar biochemical properties (polar uncharged, green; positively charged, light blue; negatively charged, purple; hydrophobic, dark blue; glycine, orange; proline, yellow). [* This alanine (A) appears in equal proportions (16%) to lysine (K)].

Of the 2,489 strains analyzed here, 1,912 are from the domain Bacteria, 444 are from the domain Eukarya, and 133 are from the domain Archaea. All 444 eukaryotic strains except one (Saccharomyces cerevisiae CLIB382, which lacks a completely annotated genome) contain at least one ANK-containing protein (Fig. 2, Table S1). 51% of bacterial strains (981/1912) and 11% of archaeal strains (15/133) harbor at least one ANK-containing protein (Fig. 2A). When strains are grouped into genera, we similarly find that 56% of bacterial genera (308/549) and 9% of archaeal genera (6/69) contain species that encode at least one protein with an ANK repeat.

Figure 2 ANK-containing protein analysis across all domains of life.

(A) Bar graph of the average percent of the strains in each domain that have one or more ANK-containing proteins. The total number of strains analyzed and the number of strains with more than one ANK-containing protein are listed below the graph. (B) Bar graph of the average number of ANK containing proteins in strains of each domain. The average number of ANK-containing proteins in each domain is listed below the graph. Error bars represent standard deviation. (*P < 0.05, ** P < 0.000001, Two-tailed Mann-Whitney U; ANOVA P < 0.000001). (C) Bar graph showing the average percent of the proteome composed of ANK-containing proteins in each domain. Error bars represent standard deviation. (*P < 0.000001, Two-tailed Mann-Whitney U; ANOVA, P < 0.000001).

For those strains with at least one ANK-containing protein, the average number and normalized percent of ANK-containing proteins per strain are shown for each major domain of life in Fig. 2B and 2C. The differences in the relative fraction of the proteome dedicated to proteins with ANK repeats are significant between the domains (Mann-Whitney U p < 0.00001).

ANK-containing proteins in bacteria

The percent of bacterial strains that contain multiple ANK-containing proteins rapidly declines as the cutoff number of ANK-containing proteins per proteome increases to four and higher (Fig. 3A). To glean which phylogenetic groups of bacteria harbor an enriched fraction of ANK-containing proteins, 24 bacterial classes spanning 202 bacterial strains encoding ≥ four predicted ANK-containing proteins were analyzed.

Figure 3 Analysis of ANK-containing proteins in Bacteria.

(A) Bar graph of the percent of bacterial strains analyzed (y axis) with the specified number of ANK-containing proteins (x axis). The number above the bars on the graph lists the number of strains with the specified number of ANK-containing proteins. (B) Consensus phylogeny of 16S rRNA sequences from one species (randomly selected) in each class. (C) Species analysis of bacterial classes that contain four or more ANK-containing protein encoding genes (only classes with 5 or more represented species were included in this analysis). The fraction in parentheses represents the number of species with four or more ANK-containing proteins in the bacterial class over the total number of species in that bacterial class.

The class with the highest fraction of ≥ four ANK-containing proteins was Sphingobacteria (Fig. 3B and 3C). To our knowledge, it is the first report that this class of typically free-living bacteria putatively encode ANK-containing proteins. Interestingly, many of the classes with a high percentage of ANK-containing proteins in Fig. 3B and 3C cluster with lineages that form symbioses with hosts, including Spirochetes, Chlamydia, and various sub-groups of Proteobacteria. As endosymbioses have independently evolved across the tree of Bacteria, the taxa are, as expected, scattered across the bacterial tree such that the relative abundance of ANK-containing proteins across the 24 classes of Bacteria is independent of phylogenetic history (p = 0.32, PI test, Reeve & Abouheif, 2003).

Enrichment of ANK-containing proteins in bacterial symbionts

To corroborate the enrichment of ANK-containing proteins in symbiotic bacteria, we categorized each taxon with four or more ANK-containing proteins into three bacterial lifestyles: (i) free-living species that solely replicate outside of host cells, (ii) facultative host-associated (intracellular and extracellular) species that can use a host for replication, and (iii) obligate intracellular species that replicate strictly within host cells. We assigned these three lifestyles following our previous annotations (Newton & Bordenstein, 2011) and searching the primary literature (Table S5).

Our comparisons reveal a striking correlation between replication strategy and abundance of proteins containing ANK repeats. Both obligate intracellular and facultative host-associated bacteria contain, on average, a significantly, higher absolute number of ANK-containing proteins than those that are free-living (Fig. 4A, Mann-Whitney U p < 0.001, ANOVA p < 0.00003), despite the notable fact that free-living species have significantly larger proteomes (Fig. 4C, Mann-Whitney U p < 0.01 for all comparisons, ANOVA p < 0.00001). Facultative host-associated strains have the most expansive repertoire of ANK-containing proteins based on absolute protein numbers (Fig. 4A and 4D), likely owing to their dual capacity to interact with eukaryotic host cells as well as retain a large genome. Consistent with these findings, a majority of the bacterial strains that contained 20 or more ANK-containing proteins are obligate intracellular or facultative host-associated microbes, while only one is free-living (Table 1).

Figure 4 Lifestyle analysis of bacterial species with four of more ANK-containing proteins.

An average of the number or percent of ANK-containing proteins for all strains of the same species was used for these analyses. FL, FHA and O denote free-living, facultative host-associated and obligate intracellular bacteria, respectively. (A) Bar graph of the average number of ANK-containing proteins in species with four of more ANK-containing proteins. Error bars represent standard deviation. (*P < 0.001, **P < 0.00001, Two-tailed Mann-Whitney U; ANOVA, P < 0.00003). (B) Bar graph of the average percent of the proteome composed of ANK-containing proteins in species with four of more ANK-containing proteins. Error bars represent standard deviation. (*P 0.001, **P < 0.0001, ***P < 0.00001, Two-tailed Mann-Whitney U; ANOVA, P < 0.00001). (C) Bar graph of the average total number of proteins in the proteomes of species with four of more ANK-containing proteins. Error bars represent standard deviation. (*P < 0.01, **P < 0.00001, ***P < 0.000001, Two-tailed Mann-Whitney U; ANOVA, P < 0.00001). (D) Bar graph of percent of species in each lifestyle that contain the specified number of ANK-containing proteins (example: 74% of obligate intracellular species, 58% of facultative host associated species, and 28% of free-living species of bacteria contain six ANK-containing proteins). (E) Bar graph of the percent of species in each lifestyle that contain the specified percent of ANK-containing proteins.

Table 1 Bacterial species with 20 or more ANK-containing proteins in our analysis.

Species	Lifestyle	Class	# ANK-containing proteins	Total Gene #	% Genes with ANK domains	Total Protein #	% Proteins with ANK domains	
Desulfomonile tiedjei DSM 6799	FL	Deltaproteobacteria	42	5664	0.742	5494	0.764	
Brachyspira hyodysenteriae WA1	FHA	Spirochaetia	60	2680	2.239	2642	2.271	
Brachyspira intermedia PWS/A	FHA	Spirochaetia	57	2926	1.948	2872	1.985	
Brachyspira murdochii DSM 12563	FHA	Spirochaetia	48	2894	1.659	2809	1.709	
Burkholderia vietnamiensis G4	FHA	Betaproteobacteria	37	7861	0.471	7617	0.486	
Brachyspira pilosicoli 95/1000	FHA	Spirochaetia	32	2336	1.370	2299	1.392	
Legionella longbeachae NSW150	FHA	Gammaproteobacteria	26	3739	0.695	3470	0.749	
Legionella pneumophila str. Paris	FHA	Gammaproteobacteria	21	3278	0.641	3166	0.663	
Turneriella parva DSM 21527	FHA	Spirochaetia	21	4214	0.498	4139	0.507	
Wolbachia sp. wPip Pel	O	Alphaproteobacteria	58	1423	4.076	1275	4.549	
Orientia tsutsugamushi str. Ikeda	O	Alphaproteobacteria	47	2005	2.344	1967	2.389	
Candidatus Amoebophilus asiaticus 5a2	O	Bacteroidetes	46	1597	2.880	1334	3.448	
Orientia tsutsugamushi str. Boryong	O	Alphaproteobacteria	37	2216	1.670	1182	3.130	
Wolbachia sp. wRi	O	Alphaproteobacteria	31	1303	2.379	1150	2.696	
Rickettsia bellii OSU 85-389	O	Alphaproteobacteria	28	1511	1.853	1475	1.898	
Rickettsia bellii RML369-C	O	Alphaproteobacteria	27	1469	1.838	1429	1.889	
Rickettsia felis URRWXCal2	O	Alphaproteobacteria	24	1551	1.547	1512	1.587	

After normalizing the dataset by the total number of proteins, the fraction of the proteome containing ANK-containing proteins is highest in obligate intracellular species (Fig. 4B and 4E). The percentage of ANK-containing proteins is inversely related to proteome size across bacterial lifestyle. In fact, a significant difference in the abundance of proteins with ANK repeats is broadly evident between the lifestyles (Mann-Whitney U p <0.001 for all comparisons, ANOVA p < 0.00001). When considering both the abundance of proteins with ANK repeats and limited proteome size, obligate intracellular bacteria have a remarkably high composition of ANK-containing proteins that not only exceeds that of other bacterial lifestyles, but also is comparable to the composition of eukaryotes in Fig. 2C.

Enrichment of repeats in ANK-containing proteins in bacterial symbionts

Obligate intracellular bacteria also harbor significantly more ANK repeats per protein (Fig. 5A; Table S6). On average, an obligate intracellular microbe contains 6.1 ANK repeats per ANK-containing protein, while free-living and facultative host-associated microbes only contain 4.6 and 4.3 ANK repeats, respectively (ANOVA p = 0.012, pairwise tests between the lifestyles, t-test p < 0.012). As discussed below, these differences likely affect protein stability.

Figure 5 Individual ANK repeat number and amino acid sequence identity analysis.

(A) Bar graph of the average number of ANK repeats in ANK-containing proteins for free-living (FL), facultative host-associated (FHA) and obligate intracellular (O) bacteria. Error bars represent standard deviation (*p = 0.0127, **p = 0.0036, t-test). For a list of strains analyzed, refer to Table S6. (B) Bar graph of the average percent of amino acid identity of the ANK repeats from the listed species with Wolbachiaw Mel ANK repeats. Strains analyzed listed in Table S8. Error bars represent standard error.

Effect of symbiont transmission on ANK-containing proteins

To determine if the mode of transmission of obligate intracellular bacteria associates with the abundance of ANK-containing proteins, we employed a previously published list of vertically and horizontally transmitted obligate intracellular bacteria (Table S7) (Newton & Bordenstein, 2011). Based on the mean of all strains from the same species (a species average), horizontally transmitted taxa (n = 24) contain more ANK-containing proteins than vertically transmitted ones (n = 6) (5.33 vs. 1.66, Mann-Whitney U p = 0.174), and have a higher percentage of their proteome dedicated to ANK-containing proteins (0.41% vs. 0.12%, Mann-Whitney U p = 0.191). However, these differences are not statistically different likely owing to the small sample size in the vertically transmitted group. If we analyze the data from each strain, the differences between horizontally (n = 31) and vertically transmitted taxa (n = 8) are marginally insignificant for the abundance of ANK-containing proteins (5.13 vs. 0.88, Mann-Whitney U p = 0.062) and proportion of ANK-containing proteins (0.39% vs. 0.11%, Mann-Whitney U p = 0.08).

ANK amino acid sequence identity across bacterial lifestyles

Two explanations for why obligate intracellular bacteria have a greater fraction of proteins with ANK repeats and ANK repeats per ANK-containing protein than facultative host-associated and free-living bacteria are: (i) ANK-containing proteins are adaptive to bacteria with an intracellular lifestyle or (ii) ANK-containing proteins experience frequent horizontal transfer between co-infecting, obligate intracellular microbes.

Fig. 5B demonstrates that there is no conservation in the ANK repeat amino acid sequence between species of the same lifestyle. For instance, when comparing the amino acid sequence of Wolbachia (strain wMel) ANK repeats to the ANK repeat sequences from other obligate intracellular, facultative host-associated and free-living microbes, there are no significant differences in the amount of sequence identity between lifestyles (Fig. 5B; Table S8). Surprisingly, Wolbachia ANK repeats are no more or less similar in sequence to each other than ANK repeats from other obligate intracellular, facultative host-associated and free-living species. Even the ANK repeat amino acid sequences of species in the same order have very little sequence identity (Fig. S1). This low level of sequence identity within and between unrelated taxa may be due to degeneracy in the ANK repeat amino acid sequence itself (Li, Mahajan & Tsai, 2006) and does not permit a demarcation of the two explanations above.

ANK-containing proteins in archaea

Of the 133 archaeal strains, 11% contain ANK-containing proteins (Fig. 2). Of these strains, the average number of ANK repeats per protein was 5.25, and four species contained more than one ANK-containing protein in their proteome (Fig. 6A). Interestingly, the ANK-containing proteins in some archaeal genera are conserved, while others are not. In the Methanosarcina genus, two species have one ANK-containing proteins with 66.9% amino acid identity. However, the three species with ANK-containing proteins from the Pyrobaculum genus have very different amino acid sequences (Fig. 6B). Other archaeal genera with ANK-containing proteins include Acidianus, Halogeometricum, Metallosphaera, Methanocella, Methanococcus, Methanothermococcus, Sulfolobus, Thermofilum, and Thermoplasma (Table S9).

Figure 6 Analysis of ANK-containing proteins in archaeal strains.

(A) Bar graph of the percent of archaeal strains analyzed with the specified number of ANK-containing proteins. The number above the bars on the graph lists the number of strains with the specified number of ANK-containing proteins. (B) Chart of the percent amino acid identity between the amino acid sequences of Pyrobaculum ANK-containing proteins.

Discussion

A central finding of this comparative study is that ANK repeats are more prevalent in bacterial species than generally recognized in the current literature, with over half of all of the 1,912 bacterial strains analyzed containing ANK-containing proteins. Far from being rare or even exclusive to certain phylogenetic groups of related bacteria, ANK repeats in Bacteria are widely distributed protein motifs. We do note that this analysis is limited to the strain information present in the SUPERFAMILY database (SUPERFAMILY). While not exhaustive, this database and our analysis spans a broad spectrum of bacterial domains, including 1912 bacterial strains, representing 992 species and 52 phylogenetic classes. Since certain strains of Bacteria that have relevance to human health naturally receive attention and have been well sampled, it is possible that the SUPERFAMILY dataset is not representative of the microbial diversity of the natural world, but rather is enriched in bacterial species that affect human health. Nonetheless, this analysis is the most comprehensive survey of ANK repeat distribution and abundance to date, leading us to conclude that previous assumptions about the rarity of ANK repeats outside of eukaryotes are exaggerated.

Evolutionary theories on the origins of the ANK repeat have evolved over time. Originally, it was assumed that prokaryotic ANK-containing proteins were obtained via horizontal gene transfer (HGT) from eukaryotic hosts, indicating that the ANK repeat originated in eukaryotic proteins (Bork, 1993). While the short sequence and divergence levels of the repeat motif between taxa precludes a clear inference of the origin of ANK repeats, there are several reasons why a single, common ancestor may be just as likely as horizontal transfer of the ANK repeat between the phylogenetic domains. First, we showed that the consensus sequences between the three domains are roughly similar, thus making it difficult to rule out that ANK repeat evolution follows the phylogeny of the domains. Second, there are several species of Archaea and non-host associated microbes that have ANK-containing proteins, which may be indicative of an older origin of the ANK repeat. Finally, although the results indicate that host-associated microbes have an increased fraction of ANK-containing proteins in comparison to free-living microbes, all lifestyles can harbor such proteins, specifying that they provide broader advantages to the cell. Whether or not these proteins were inherited by HGT or evolved by descent with modification from a common ancestor, the distribution for these proteins in Bacteria and Archaea has been unknown and warrants functional and evolutionary analyses.

While ANK repeats in eukaryotes are ubiquitous structural motifs that facilitate a myriad of protein-protein interactions, our analysis reveals that ANK repeats cluster to some degree in symbiotic bacteria involved in microbial-host interactions. Recent studies of host-associated bacterial species, including, Legionella pneumophila (Al-Khodor et al., 2010; de Felipe et al., 2008), Anaplasma phagocytophilum (IJdo, Carlson & Kennedy, 2007), and Ehrlichia chaffeenis (Zhu et al., 2009), show that ANK-containing proteins can be secreted through a type IV secretion system into the cytoplasm of their host and alter host gene expression and interfere with its hosts’ microtubule directed vesicular transport, respectively (Garcia-Garcia et al., 2009; Pan et al., 2008; Zhu et al., 2009). Based on our data, bacterial ANK-containing proteins may play a significant role in ensuring the pathogen’s survival within the host cell.

Protein folding studies indicate that higher numbers of ANK repeats in a protein results in increased structural stability (Hagai et al., 2012; Mello et al., 2005; Wetzel et al., 2008). We observed that obligate intracellular microbes, on average, have 6.1 ANK repeats per protein, in comparison to 4.6 and 4.55 in bacteria with free-living and facultative host associated replication, respectively (Fig. 5A). This significant difference suggests that the proteins with ANK repeats in obligate intracellular bacteria have a more stable structure than those from bacteria in the other two lifestyles. Furthermore, a study on the folding dynamics and stability of DARPins (designed ankyrin repeat proteins) composed of identical ANK repeats designed from a consensus ANK repeat found that when the number of ANK repeats was reduced from 7 to 4, the stability of the protein was substantially reduced (Wetzel et al., 2008). Coincidentally, this difference in the number of ANK repeats is similar to that observed between obligate intracellular bacteria and free-living/facultative host associated lifestyles in our analysis. Taken together, we suggest that the ANK-containing proteins in obligate intracellular species have, on average, a more stable structure that could potentially underlie more effective interactions between bacterial effector proteins and host proteins. Interestingly, recent proteomic evidence has indicated that some obligate intracellular bacteria, including Blochmonnia chromaiodes and Buchnera, express an abundance of chaperones, such as GroEL, in an effort to provide greater stability for proteins that have accumulated deleterious mutations (Fan et al., 2013; Poliakov et al., 2011). It is possible that enhanced stability of the ANK domain conferred by the accumulation of additional ANK repeats is not required to provide stability for protein interactions, but is rather part of an overall effort to increase protein stability.

On a related note, a comparative study on ANK domain-encoding genes (ANK genes) present in species of Wolbachia pipientis that inhabit Drosophila found that these ANK genes are rapidly evolving due to homologous and illegitimate recombination via the short direct repeat sequences (Siozios et al., 2013). The authors speculated that since stress-related genes also contain these types of direct repeats, which allows for rapid change in challenging environmental conditions, ANK-containing proteins may be used in similar stressful conditions such as directly interacting with host tissues or proteins (Rocha, Matic & Taddei, 2002; Siozios et al., 2013). This inference complements the findings of our analysis because the enriched repertoire of ANK-containing proteins and ANK repeats per protein in obligate bacteria may aid intimate host-microbe interactions.

A number of pathogenic microbes that contain ANK-containing proteins have been identified in this study. For instance, the microbe with the greatest number is the spirochete, Brachyspira hyodysenteriae,which remarkably has 60 ANK-containing proteins. B. hyodysenteriae is a classic gastrointestinal pathogen and the causative agent of a wide range of diarrheal diseases in pigs that naturally leads to significant economic ramifications (ter Huurne & Gaastra, 1995). Of B. hyodysenteriae’s 60 ANK-containing proteins, 34 contain a signal sequence for secretion (Table S10) suggesting that many of these proteins, if expressed, are exported from the microbe into its host that may facilitate pathogenesis (Bellgard et al., 2009; Mappley et al., 2012).

The number of ANK-containing proteins within a group of closely related taxa can be extremely variable. In the order Campylobacterales, Helicobacter hepaticus has 13 such proteins, Helicobacter mustelae has two proteins and Helicobacter cinaedi has three. The remaining five Helicobacter species in our analysis do not have any ANK-containing proteins (Table S11). The related Campylobacter species, including Campylobacter jejuni, have two to three (Table S11), and some ANK-containing proteins in Helicobacter and Campylobacter are probable orthologs (Fig. S2). Interestingly, one ANK-containing protein present in both H. cinaedi and C. jejuni is required for C. jejuni colonization due to its capacity to reduce levels of reactive oxygen species (ROS) in the cell (Flint, Sun & Stintzi, 2012). Finally, the increased repertoire of ANK-containing proteins in H. hepaticus, particularly the three proteins with secretion signal sequence and the two proteins with transmembrane domains (Table S12), may associate with this species’ unique infection of the lower bowel and liver of its host, resulting in inflammatory bowel disease, chronic hepatitis, and liver cancer (Suerbaum et al., 2003).

Although the vast majority of the species with the highest number of ANK-containing proteins are host associated, Desulfomonile tiedjei is an outlier because it harbors 42 such proteins (Table 1). D. teidjei is an anaerobic, free-living bacteria that dechlorinates hydrocarbons, such as tetrachloroethylene (PCE) and trichloroethylene (TCE) (Deweerd & Suflita, 1990). The fact that D. tiedjei also harbors 42 ANK-containing proteins, 19 of which also contain signal sequences, has, to our knowledge, not been reported nor discussed in this microbe’s bioremediation capabilities (Table S13). Although it dechlorinates PCE and TCE, D. teidjei cannot use these chemicals as a carbon source. Instead, D. teidjei lives syntrophically with other anaerobic microbes and relies on them for nutrients (Shelton & Tiedje, 1984). We speculate based on widespread enrichment of ANK-containing proteins in symbionts that these ANK-containing proteins could play a role in this interaction.

Conclusions

Our analysis of the ANK protein motif, augmented with the taxon lifestyles and phylogeny, upgrades the magnitude of ANK repeat biology across the diversity of life. The enrichment of ANK-containing proteins in host-associated bacteria signifies that they are not evolutionarily restricted to unique types of Bacteria or Archaea, but instead can independently thrive in diverse taxa. The functional roles of ANK-containing proteins in Bacteria and Archaea remain understudied and will be an exciting frontier for future investigations of protein interactions between the different domains of life.

Supplemental Information

Supplemental Information 1 Raw data extracted from SUPERFAMILY for Domain analysis.

Click here for additional data file.

Supplemental Information 2 ANK-containing proteins used to generate the Eukarya ANK repeat consensus sequence.

Click here for additional data file.

Supplemental Information 3 16S sequences selected to construct the bacterial class phylogenetic tree.

Click here for additional data file.

Supplemental Information 4 Frequency of the consensus amino acid at each position in the consensus sequence for Eukarya and Archaea ANK repeat.

Click here for additional data file.

Supplemental Information 5 Data for Lifestyle analysis.

Click here for additional data file.

Supplemental Information 6 Genomes used for ANK repeat number analysis.

Click here for additional data file.

Supplemental Information 7 Genomes used for transmission analysis.

Click here for additional data file.

Supplemental Information 8 Genomes used for ANK repeat percent identity analysis.

Click here for additional data file.

Supplemental Information 9 Ankyrin amino acid sequence identity: Comparison between species of the same order.

All ANK repeat sequences from the following species were analyzed: Legionella pneumophila Philadelphia 1, Coxiella burnetii Dugway 7E9-12, Wolbachia pipientis wMel, Ralstonia solanacearum PSI07, Burkholderia vietnamiensis G4, Leptospira biflexa serovar Patoc strain ’Patoc 1 (Paris), Treponema pallidum pallidum Nichols. To analyze the individual ANK repeats, the ANK-containing proteins sequence ID was obtained from SUPERFAMILY v1.75 and the amino acid sequence information was obtained from NCBIs Proteins database (http://www.ncbi.nlm.nih.gov/protein/). SMART (http://smart.embl-heidelberg.de/) was used to identify the number and location of each individual ANK repeat in the ANK-containing protein. For the amino acid sequence identity analysis, individual ANK repeat sequences were aligned using MUSCLE and the percent identity of the sequences was calculated in Geneious Pro 5.6.2. When comparing ANK repeat sequences from two strains, the average of all combinations of ANK repeat comparisons was used. All species ANK repeat sequences were compared to Wolbachia pipientis wMel to show that the level of identity was the same between species of the same order, and that of a different order (Wolbachia).

Click here for additional data file.

Supplemental Information 10 Archaea species with ANK-containing proteins.

Click here for additional data file.

Supplemental Information 11 ANK-containing proteins in Brachyspira hyodysenteriae.

TM, transmemebrane domain. The number in the parentheses in the TM domain column refers to the number of TM domains the protein is predicted to have by SMART.

Click here for additional data file.

Supplemental Information 12 Campylobacter species analyzed.

Click here for additional data file.

Supplemental Information 13 Phylogeny of ANK domains of ANK-containing proteins in Helicobacter hepaticus, Helicobacter cidaedi, and C jejuni.

SMART was used to identify ANK repeats, and the ANK domain contained all ANK repeats and some internal linker sequences. MUSCLE alignment was used to align amino acid sequences. INDELS and ends of amino acid sequences were removed after being aligned. Geneious builder was used to build the Neighbor-Joining tree. Blue boxes; Helicobacter hepaticus & Helicobacter cidaedi orthologs Red Box: Helicobacter hepaticus, Helicobacter cidaedi, and C. jejuni orthologs.

Click here for additional data file.

Supplemental Information 14 ANK-containing proteins in Helicobacter hepaticus.

TM, transmembrane domain. The number in the parentheses in the TM domain column refers to the number of TM domains the protein is predicted to have by SMART.

Click here for additional data file.

Supplemental Information 15 ANK-containing proteins in Desulfomonile tiedjei.

TM, transmemebrane domain. The number in the parentheses in the TM domain column refers to the number of TM domains the protein is predicted to have by SMART. * Desti_0449 has a HPT (Histidine Phosphotransfer), a REC (cheY-homologous receiver) and a GGDEF(diguanylate cyclase) domain. **Desti_1504 has a Peptidase C14 (Caspase domain) domain.

Click here for additional data file.

Additional Information and Declarations

Competing Interests

Author Contributions

The authors declare that they have no competing interests.

Kristin Jernigan conceived and designed the experiments, performed the experiments, analyzed the data, wrote the paper.

Seth R Bordenstein conceived and designed the experiments, analyzed the data, wrote the paper.

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
