# Peer review of "Ankyrin domains across the Tree of Life"

_PeerJ, doi:10.7717/peerj.264_

## Round 0.1 · original submission · Major Revisions

The reviewers have all recommended revision of the paper and are unanimous in some of their criticisms, which need to be carefully and fully addressed. I would also ask you specifically to consider the following points:

1. The paper’s results have been over-interpreted, by which I mean more critical consideration is needed to justify the conclusions. For example, the limitations of the SUPERFAMILY database (on which the results are based) should be considered – it contains only about 1200 prokaryotic genomes. Thus, a little more circumspection is needed, recognising that, although the results are of interest, they are, in many respects, far more preliminary than the paper suggests.

2. The presence of ANK repeats in prokaryotes obviously provides an additional perspective on potential structure-function relationships. However, the results in the present paper point only to correlations and not to causal associations – this point is not made sufficiently clear. The correlation with “lifestyle” is problematic and needs to be better argued (e.g. as noted by one of the reviewers, the argument that D. tiedjei contains a large number of ANK proteins because it is syntrophic is far too speculative – what about other syntrophs???)

3. Use of terminology also needs clarification throughout the paper. Most importantly, ANK is used interchangeably to refer to repeats, domains and proteins. Reviewers have also highlighted that taxonomic terminology is confusing and, in several cases, wrong and misleading.

4. The paper’s format is not suitable for publication. For example, different font style and sizes are mixed, and the reference citations in the body of the paper and bibliography do not conform to the required style.

·

Basic reporting

The article is fine.


Title
In a paper on the 'tree of life" I would expect also viruses even if viruses don't live. If you compare different domains of life, especially in relation to being facultative or obligate intracellular, viruses become very interesting. Viruses do have ANK proteins but they are sadly missing here.

Abstract
... oppose the concept that ...
I would like to suggest rephrasing the statement less as an opposition. It is running in open doors. The senior author is a well-established member of the symbiosis field and he always like everyone else in the field saw ANK repeats as a hallmark of host-symbiont interactions.


Figure 4 D and E.
I struggle to understand the concept of the percentages. A more explanatory legend might help, perhaps even including an example.

The authors might wish to discuss the recent paper by Siozios et al. on diversity and evolution of Wolbachia ankyrin repeat domain genes.


Minor points

1st line Abstract ... repeats are ...
3rd line Abstract, L41 bacteria without capital or in italics with capital
L27 33-amino acid ...
L62-64, L82-83 formatting
Species names in references should be in italic.
The page numbers for Suraj Sing are now available.

Experimental design

The experimental design is fine

Validity of the findings

The validity of the findings is fine with two exceptions.

Figure 1
This is a step backwards.
The figure shows the comparison of three sequences, Eukarya, Archaea and Bacteria. However, the first sequence is Eukarya and Bacteria and the whole comparison becomes muddled. This is being lazy. All the ANK people in the field are still cutting this corner and this paper could make a real difference by, for the first time, reporting a true or clean consensus sequence for Eukarya.

Figure 2, 4 and 5A
The standard errors are wrong, it should be standard deviations.

Additional comments

A new consensus for Eukarya??

Reviewer 2 ·

Basic reporting

I enjoyed reading this manuscript. It is well written, well researched, and well organized. Its premise is simple but creative and it makes an important contribution to the literature.

Its first contribution is to review available data to point out that ANK repeats are widespread and common among Bacteria and Archaea and so perhaps should no longer be thought of as "eukaryotic motifs". This in and of itself is a useful observation.

The second hypothesis proposed by the authors is that the occurrence of ANK-repeat proteins correlates more strongly with lifestyle than with phylogeny. The lifestyles are defined as free-living, obligate symbiont, and facultative symbiont. In the examined dataset ANK repeat proteins are numerically most abundant in facultative symbionts but comprise a greater proportion of predicted proteins in obligate symbionts. The authors then propose that this indicates a likely role for the ANK repeat proteins in symbiont-host interaction.

Experimental design

This work is based on existing literature and published data. Treatment of these data appears to be thorough and appropriate, with appropriate statistics applied where necessary.

Validity of the findings

This last claim (see Basic Reporting above) is the weakest in my opinion. Or perhaps I should say that this conclusion may be too broadly drawn.

First, while the study looks comprehensively at ~2500 available genomes represented in SUPERFAMILY, it is not clear to me how well these represent the broad diversity of bacteria and archaea, or more importantly how this selection of genomes may be biased toward certain types of organisms. For example, Table 1 appears to show that the organisms with the largest number of ANK proteins fall into two distinct categories, “facultative symbionts” that mainly infect mammals and “obligate symbionts” that mainly infect arthropods or arthropods and mammals. This is a rather narrow representation of the huge diversity of bacteria and eukaryotes involved in symbiotic interactions. So my sense is that there is a danger here of overgeneralization.

Secondly, the authors point out that another possible role for the ANK repeats is protein stabilization. Many obligate endosymbionts over-express molecular chaparones, which may be an indication that protein stability is a problem for obligate intracellular symbionts. This could suggest that the observed correlation is not due to a direct role of ANK proteins in mediating host symbiont interactions, but instead might be a secondary response to a set of conditions that may be experienced by symbiotic microorganisms, but may not necessarily be unique to the symbiotic lifestyle. This could explain why free-living organisms, such as Desulfomonile tiedjei also encode many ANK proteins. Perhaps byproducts of their metabolism or environment create a need for more stable proteins.

I would like to see a more nuanced discussion that mentions other possible explanations for the apparent correlation of ANK frequency in genomes and symbiotic lifestyle. I also would like to see some discussion of the extent to which the selection of symbioses investigated represents the broad lifestyle definitions proposed here or whether the authors should attempt a more narrowly circumscribed hypothesis, i.e. maybe these observations are more correlated with life in mammalian or arthropod hosts rather than with symbioses across the entire tree of life.

Also, a few specific comments:

The Line 55: Meaning unclear...missing word or punctuation? "We note appropriate caution that ANKs are called based on a computational framework and are not experimentally confirmed." Or perhaps you meant to use the word "identified" rather than "called" which can be construed as slang or jargon.

Lines 168-171: These data indicate that among the examined organisms ANK proteins are more frequent/protein in obligate symbionts, but has the sequence space for symbionts really been adequately sampled by this selection of taxa? Is this an overgeneralization?

Line 221: This seems like an overly broad conclusion based on the presented data.

268 Syntrophically not symtrophically.

Line 268-271. I think that this statement is a real stretch. It is too broad and does not refer to any plausible mechanism for the proposed causal relationship.

Additional comments

Don't get me wrong. I truly enjoyed reading this paper. I found it simple but thought provoking and potentially important. I think that you could well be correct in your broad conclusions. I just don't believe that you have proved them. It is important to be conservative, because as we have all seen, speculations can sometimes become entrenched in the literature as fact, unless they are clearly identified as speculations by the authors.

Reviewer 3 ·

Basic reporting

Basic reporting:
To the best of my understanding this submission:
• -Does adhere to PeerJ policies
• -Is clear in most sections. The following are comments/suggestions the authors may consider:
-Line 21: Not clear what “(p16)” refers to in the sentence. Is this part of the reference by Tang, Fersht et al.?
-In general, figures with results should not be included/referred in the Material and Methods section. If they do, then the numbering of figures and tables should reflect their order of appearance in the manuscript. For example, Table S2 is mentioned before Table S1 (line 58) and Fig. 3 is referred to before Figs. 1 and 2 (line 78)
-Lines 62-64 have different font size than the rest of the manuscript.
-Line 65: Not clear what “the individual ANK repeats” refers to. Reappears in lines 132, 202.
-Line 79-81: Not clear what was considered to select the DNA substitution model for alignment in jModel Test.
-Line 81: Period missing.
-Line 83: Usage of “ANKs” without definition. Not clear is ANKs refers to ankyrin repeats, domains, or proteins.
-Line 91: change “Life” to life.
-In general, in the Results and Discussion sections the authors often mention the word “organism(s)”. It is not clear if in some/all instances this refers to the taxonomic species, to strains within one species, to individual organism within a taxonomic group, or other. Please be more specific. Examples include lines 100, 101, 107, 108, 113, 116, 194, etc.
-Line 11: Refer readers to “section ANK Protein in Bacterial Symbiosis”.
-Line 116: Incorrect use of “interrogated”.
-Please rephrase line 158.
-Line 193: “finding from these results” could be instead “….from this study (ies).
-Fig. 1: please add a description of the color scheme used in the amino acid sequence, i.e. yellow means …., green …., etc.
-Fig. 2: As in earlier comment, it’s not clear exactly what the word “organisms’ refer to, species? Also, use word “Two” in Two-tailed, instead of number 2.
-Fig. 3: This legend needs clarification. As above, “organisms”? Please refer to axis X when talking about 3A. Panel B could benefit from braking into B and C to avoid using “(On the left)” and “(On the right)”. Consensus phylogeny? Or phylogeny of consensus? Four, instead of “4”.
-Fig. 4: “organisms” refers to strains with in one species? Two-tailed instead of “2-tailed”.
-Fig. 5: Legends is not clear. Incomplete sentences when referring to free-living, facultative…. species, groups? For a list of species analyzed refer to Table S4…
-Table 1: legend would benefit with the addition of a brief description of the analysis.

Experimental design

To the best of my understanding this submission:
• This submission describes original work, has been properly conducted.
• The following comments should be considered by the authors to clarify methodology:
-Line 78: Not clearly stated the reason to build a phylogenetic tree using only one “randomly” selected species and the use it to relate to abundance of ANK in those phylogenetic groups.
-Line 79-81: Not clear what was considered to select the DNA substitution model for alignment in jModel Test.
-In similar fashion, few details are provided on the parameters used during the various analyses in the different software products to identify ANK sequences, their alignments, and phylogenetic independence. More information is needed for clarification and potential reproducibility.

Validity of the findings

To the best of my understanding this submission:
-The data used for analyses in these studies is available to the readers in the manuscript or in included supplementary files.
-Conclusions are appropriate.
-No further comments in this section.

Additional comments

-I found this manuscript interesting and informative. The observation that ankyrin-containing proteins may be involved in lifestyle of prokaryotes is a novel topic and may benefit many different research areas including those of pathogenesis and mutualism. The manuscript opens novel areas of study for those interested in the evolution of molecular motifs/domains and their functions across phyla.
-Care must be placed to clarify distinctions between taxonomic groups throughout the manuscript. I was often confused with the use of “organism” and “species”, and not sure when they were interchangeable and when not.
-Another source of confusion was the use of the acronym ANK, sometimes referring to ankyrin repeats, domains, and proteins.

---

## Round 0.2 · Minor Revisions

Thank you for attending to all of the issues and points raised by the reviewers so comprehensively, and please accept my apologies for any inconvenience caused by the delay in making a decision on the revised paper.

Please make sure the the final version of the paper for publication is carefully checked with respect to format changes. For example, some references are incorrectly listed in the bibliography: Deweerd & Sulfita (1990), article title should be in sentence case; Singh et al. (2013), reference incomplete.

---

## Round 0.3 · accepted · Accept

Thank you for the revised paper, which is now acceptable for publication.